# Brief communication: Using averaged soil moisture estimates to improve the performances of a regional scale landslide early warning system

Samuele Segoni[1], Ascanio Rosi[1], Daniela Lagomarsino[1,2], Riccardo Fanti[1], Nicola Casagli[1]

[1] Department of Earth Sciences, University of Florence, Firenze, 50121, Italy
[2] Now at ENI s.p.a.

*Correspondence to*: Samuele Segoni (Samuele.segoni@unifi.it)

**Abstract.** We communicate the results of a preliminary investigation aimed at improving a state-of-art RSLEWS (regional scale landslide early warning system) based on rainfall thresholds by integrating mean soil moisture values averaged over the territorial units of the system. We tested two approaches. The simplest can be easily applied to improve other RSLEWS: it is based on a soil moisture threshold value under which rainfall thresholds are not used because landslides are not expected to occur. Another approach deeply modifies the original RSLEWS: thresholds based on antecedent rainfall accumulated over long periods are substituted with soil moisture thresholds. A back analysis demonstrated that both approaches consistently reduced false alarms, while the second approach reduced missed alarms as well.

## 1 Introduction

Regional scale landslide early warning systems (RSLEWS henceforth) are usually based on empirical rainfall thresholds, which in turn are based on rainfall parameters that can be easily measured and monitored by rain gauges (Aleotti, 2004; Baum et al., 2010; Cannon et al., 2011; Segoni et al., 2015a; Leonarduzzi et al., 2017; Piciullo et al., 2017).

However, it is widely recognized that soil moisture conditions before the triggering rainfall event can play a crucial role in the initiation of landslides, especially if deep-seated landslides and terrains with complex hydrological settings are involved (Wieczorek, 1996; Zezere et al., 2005; Jemec and Komac, 2013; Peres and Cancelliere, 2016; Bogaard and Greco, 2017).

Unfortunately, the influence of soil moisture conditions is difficult to be encompassed into RSLEWS. One of the most widespread approaches is establishing rainfall thresholds based on the rainfall amount accumulated during a given period before landslide occurrence or before the triggering rainfall event (Kim et al., 1991; Chleborad, 2003). The length of these timespans varies widely in the international literature, e.g. from a few days (Kim et al., 1991; Calvello et al., 2015) to a few months (Zezere et al., 2005). More advanced models combine daily rainfall data to compute antecedent rainfall indexes that can be used to forecast landslide occurrence (Crozier, 1999; Glade et al., 2000). All these methodologies share the approach of considering antecedent rainfall as a proxy for soil moisture. A smaller series of studies takes advantage of remotely sensed

soil moisture data (Brocca et al., 2015; Laiolo et al., 2015) but their integration in RSLEWS is not straightforward and it is limited to few case studies (Ponziani et al., 2012).

This work explores the possibility to exploit the estimated mean soil moisture value averaged over large (thousands of squared kilometers) territorial units to find an empirical correlation with the triggering of landslides.

5 We tested this hypothesis in the regional warning system of the Emilia Romagna Region (Italy), which is based on the combination of short-term and long-term rainfall measures to forecast the occurrence of landslides, as described in detail in Martelloni et al. (2012) and Lagomarsino et al. (2013). We developed an alternate version of the RSLEWS, substituting long term measures with soil moisture estimates obtained by TOPKAPI, a physically based model (Ciarapica and Todini, 2002). The different versions of the RSLEWS were compared and, given the satisfactory outcomes of the results, we discuss a possible

10 application of the proposed methodology to the regional warning system.

## 2 Materials and method

Test site is the Emilia Romagna Region (Northern Italy). This region is characterized by a morphology ranging from high mountains in the S-SW to wide plains towards NE. The mountain chain of the region belongs to the Northern Apennines, which is a complex fold-and-thrust arcuate orogenic belt originated in response to the closure of the Ligurian Ocean and the

15 subsequent collision of the European and continental margins which started in the Oligocene (Agostini et al., 2013). The mountainous part of the region is affected by surficial and deep-seated landslides, which can be triggered by short and intense rainfalls or by prolonged rainy periods, respectively (Martelloni et al., 2012).

One of the instruments used to manage landslide hazard is a RSLEWS called SIGMA, which is based on a complex decisional algorithm considering the overcoming of statistical rainfall thresholds (Martelloni et al. 2012). The thresholds are defined in

20 terms of standard deviation ($\sigma$) from the mean rainfall amount accumulated during progressively increasing time steps.

The methodology to develop sigma model (fully described in Martelloni et al. 2012) is based on the hypothesis that anomalous or extreme values of rainfall are responsible for landslide triggering and multiples of the standard deviation ($\sigma$) are used as thresholds to discriminate between ordinary and extraordinary rainfall events. To obtain probability values of not exceeding a given rainfall threshold, rainfall time series longer than 50 years are taken into account for each rain gauge. Data of the original

25 rainfall distributions are adapted to a target function chosen as a model (Standard Gaussian distribution in this case). After this conversion, it is possible to define any probability of not overcoming by using standard deviation values, which in turn can be related to the corresponding rainfall value of the original data series.

SIGMA algorithm considers two different periods of cumulative rainfall. Daily checks of 1, 2 and 3-day cumulative rainfall (short period) are used to forecast shallow landslides. A series of daily checks over a longer and variable time window (ranging

30 from 4 to 243 days depending on the seasonality) is used to forecast deep seated landslides in low-permeability terrains (Lagomarsino et al. 2013). To increase the effectiveness of the model, the mountainous part of the region is divided into 25

homogeneous territorial units (TU), each monitored by a reference rain gauge, as fully described in Lagomarsino et al. (2013) and depicted in Figure 1.

For some of the hydrographic basins of the region, ARPAE-ER (Regional Agency for Prevention, Environment and Energy of Emilia Romagna) provides the mean soil moisture value at hourly time step. These values are estimated by TOPKAPI
(TOPographic Kinematic APproximation and Integration) (Ciarapica and Todini, 2002), which is a rainfall-runoff model providing high-resolution hydrological information.

We used these data to estimate the mean daily soil moisture (MSM) value for each TU. We used daily aggregation because SIGMA is normally run daily, and it uses daily aggregations of hourly rainfall measurements; therefore, a higher temporal resolution would be unnecessary. In case the territory of some TUs is occupied by more than one basin, a weighted mean was
used to obtain an averaged value.

Similarly, since the final objective of this work is coupling soil moisture data with rainfall data measured over discrete points (a network of rain gauges, one for each TU), we are not interested into distributed modeling of soil moisture, but a single soil moisture value is needed for each TU. This approach is not completely new, as in the same test site Martelloni et al. (2013) used punctual measurements of temperature to incorporate in SIGMA a module accounting for snow accumulation/depletion
processes.

## 3 Alternate approaches

### 3.1 A preliminary test: the mean soil moisture (MSM) threshold

We compared all landslide occurrences in the years 2009-2014 and MSM (mean soil moisture) at each TU. We verified that for each TU a threshold MSM value can be identified under which landslides have never been reported, independently from
the rainfall amount. In addition, we verified that in general TUs had similar threshold MSM, with a few exceptions. Threshold MSM is 75% in TU23 and TU22, 76% in TU18, 78% in TU17, and 79% in TU19. In TU21, the threshold MSM is 88%. This value is higher than all other TUs and it can be partially explained with the scarcity of data: only 4 landslide events are included in the testing dataset of TU21. TU20 presents a landslide event with 54% MSM. If we consider this event as an outlier and we exclude it from the analysis, the value is 75% also for TU20.
Consequently, taking a MSM threshold into account could prevent SIGMA from committing false alarms in case of abundant rainfalls outside the rainy season, when the soil is dry. Therefore, we modified SIGMA algorithm adding a cut-off threshold defined as MSM = 75%, which is the arithmetic mean of the values of each TU. Basically, the modified version of the algorithm checks the daily MSM value reported for a given TU, and compares it with the MSM=75% threshold. Under this value, no landslide is expected and the SIGMA algorithm is not launched. If daily MSM is higher than 75%, landslides could be expected
if particular rainfall conditions are verified, therefore SIGMA algorithm is launched. We set a MSM threshold equal for all TUs because in some TUs the landslide dataset contains only a few events (e.g. only 4 landslide events in TU21) and a dedicated MSM threshold value would be characterized by a very weak empirical correlation that would prevent a safe use in

the RSLEWS. In addition, if we exclude the outliers, all TUs are characterized by small variations in MSM threshold values (from 75% to 79%). We therefore decided to renounce at the "detail" of the personalized threshold in favor of a more robust MSM threshold generalized for the whole test area.

A back analysis performed for the years 2009-2014 over the 7 test TUs shows a marked reduction of false alarms (days in which the rainfall thresholds are exceeded but no landslides are reported). More in detail: false alarms in the first warning level decreased from 320 to 231 (-28%), false alarms in the second warning level decreased from 169 to 141 (-17%) and false alarms in the third warning level decreased from 13 to 5 (-62%). To correctly evaluate the effectiveness of a EWS, the improvement concerning false alarms should be weighed against the behavior concerning missed alarms (days in which the rainfall thresholds are not exceeded but landslides are reported). We verified that the introduction of the MSM threshold caused the increase of missed alarm counts only by 1: the already mentioned event occurred in 01/06/2013, consisting in three landslides (lowest alarm level according to Lagomarsino et al., 2013). Since this was a minor event and since lowering the MSM threshold to 54% would result in an almost total loss of the benefits in terms of false alarm reduction, the 75% threshold was considered successfully tested and the 01/06/2013 event was considered an acceptable trade-off for a general improvement of the warning system.

It should be noted that the described use of the MSM threshold is not capable of reducing the missed alarms committed by SIGMA, as it acts like a cut-off filter. To obtain a reduction of both missed and false alarms, a more radical modification of SIGMA is depicted in the next section.

**3.2 SIGMA-U**

After the preliminary but encouraging results described in the previous section, we decided to integrate soil moisture thresholds more deeply into the original SIGMA algorithm, and we substituted rainfall thresholds based on long accumulation periods with statistical soil moisture thresholds. Following the same procedure used in Martelloni et al. (2012) for rainfall data to build σ curves, we calculated for every TU the time series of soil moisture (u), assessing the mean values and the standard deviations. After this procedure, for each TU every soil moisture value (U) could be expressed in terms of multiples of standard deviation from u.

After that, we deeply modified the original decisional algorithm of SIGMA, discarding all the long-period rainfall σ curves in favor of soil moisture σ curves. While the former rainfall σ curves were checked for long periods up to 243 days, the new soil moisture σ curves are checked for cumulative periods ranging from 1 day to 15 days, at 1 day increasing time steps. Rainfall thresholds based on rainfall sigma curves are still present in the new version of the algorithm, but are used only for short periods (1 day, 2 days and 3 days antecedent rainfall). The new version of the algorithm, which was called SIGMA-U, is shown in Fig. 2.

A back analysis was performed using landslide, soil moisture and rainfall data of the period 2011-2014 to compare the performances of SIGMA and SIGMA-U. The test was performed in all TUs where soil moisture values were available (14 out of 25, as shown in Figure 1) and the results are summarized in table 1.

The results of the back-analysis are encouraging, as the count of both false alarms and missed alarms is lower in SIGMA-U than in SIGMA. Concerning false alarms, the more dangerous the alarm level, the higher the reduction: false alarms corresponding to the 1st warning level, which are negligible, decreased by 8%, while the very important warning level 3 was erroneously issued 11 times instead of 21 (-48%). False alarms at the intermediate warning level 2 were reduced from 287 to
197 (-31%). Missed alarms were reduced as well: while SIGMA missed 88 alarms, SIGMA-U missed 69 alarms (-22%). This corresponds to a total of 134 missed landslides instead of 214 (-37%). Overall, SIGMA-U hits 789 landslides out of 923 (85.5%), outperforming SIGMA that hits 709 landslides (76.8%).

## 4 CONCLUSION

We communicate the results of a preliminary investigation aimed at improving a state of the art RSLEWS based on rainfall thresholds (SIGMA, Martelloni et al., 2012; Lagomarsino et al., 2013) by integrating mean soil moisture values averaged over the territorial units of the system. We tested two different approaches. The first approach is the simplest: it is based on a soil moisture threshold value (75% in this study) under which rainfall thresholds are not used because landslides are not expected to occur. When tested with a back analysis, this approach reduced consistently false alarms, but produced an additional missed
alarm. This approach is very simple and can be easily replicated in other cases of study after a straightforward calibration against the local soil moisture and landslide datasets.

The second approach is more complex and relies on the idea that rainfall thresholds based on antecedent rainfall accumulated over very long periods can be substituted with soil moisture thresholds. A back-analysis demonstrated that a new version of the model based on soil moisture and short-term rainfall could be more effective than the original version based on short-term
rainfall and long-term rainfall, as both false alarms and missed alarms were consistently reduced.

Some recent studies criticized the traditional rainfall threshold approach based only on rainfall variables, and stressed the importance of considering additional factors such as soil moisture to better encompass the hydrologic conditions of landsliding slopes (Bogaard and Greco, 2017; Canli et al., 2017). The present work follows the direction expressed by the aforementioned series of works and presents a small advance towards a sounder (and more effective) hydrologic approach to identify rainfall
thresholds for landslide occurrence.

The research is still ongoing and further tests are needed before arriving to a full integration with the regional landslide warning system of Emilia Romagna. These tests include: (i) the use of soil moisture measurements coming from other sources (e.g. remotely sensed data or direct measurements at selected test sites); (ii) the refinement of the spatial resolution of the alerts by integrating soil moisture measurements, rainfall thresholds and susceptibility maps (Segoni et al., 2015b); (iii) the improvement
of the model taking into account different threshold values of sigma for each TU, after a thorough site-specific calibration; (iv) a thorough validation of the model.

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

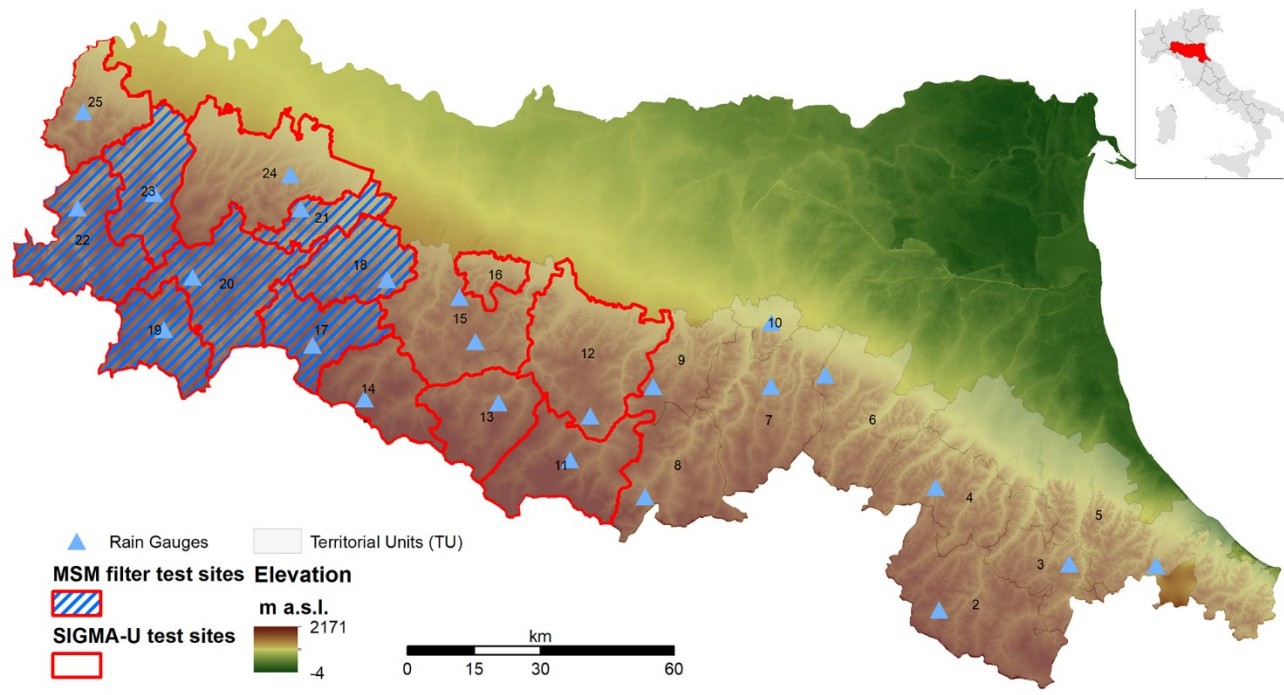

**Figure 1: Test site showing the partition in Territorial Units (TU) and highlighting the TUs used as test sites.**

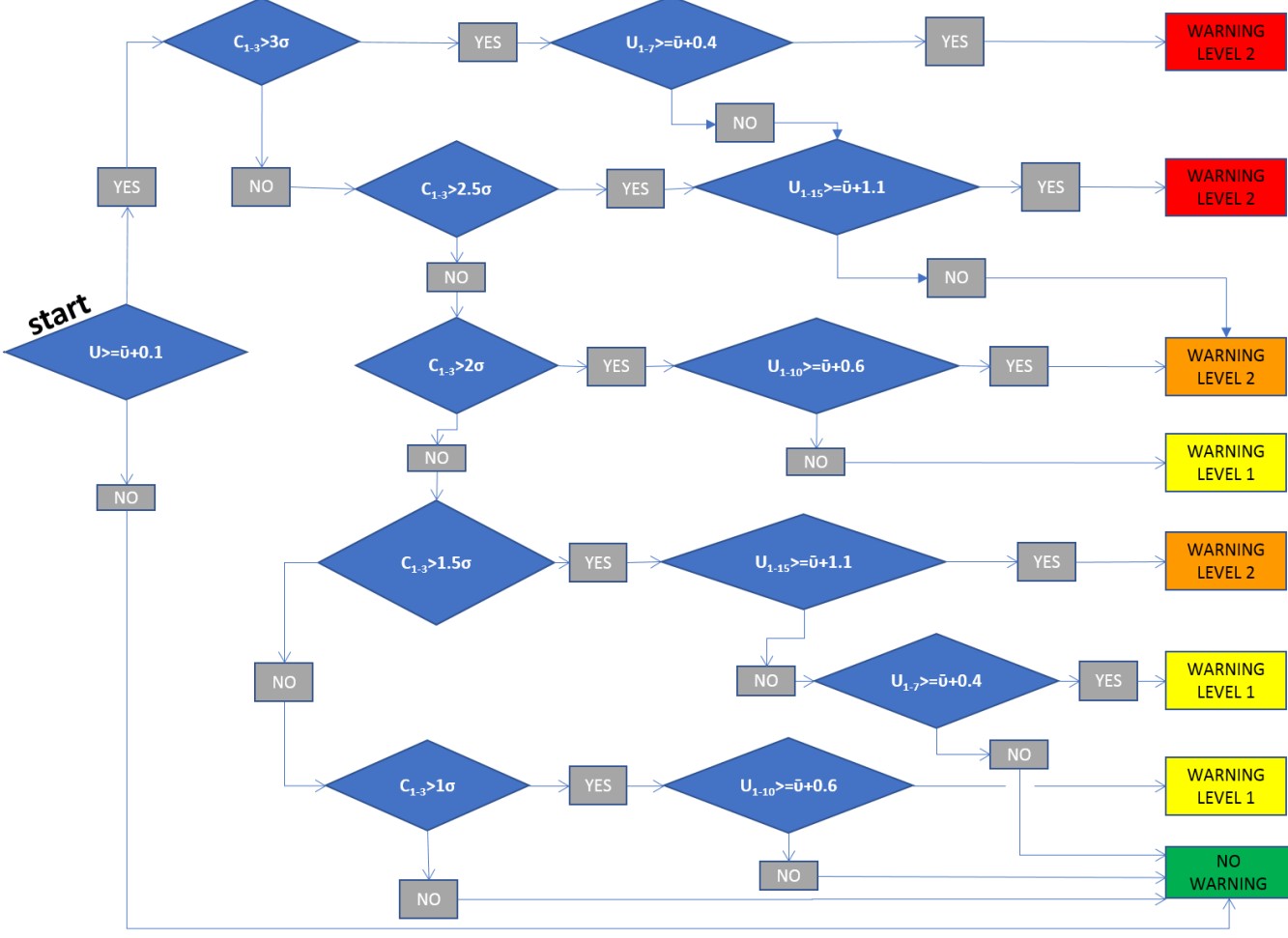

**Figure 2: Scheme of the SIGMA-U algorithm. C: cumulative rainfall, U: soil moisture, ū: average soil moisture.**

**Table 1: Quantitative evaluation of the performances of the models SIGMA (Lagomarsino et al., 2013) and SIGMA-U (this paper).**

| | | SIGMA | SIGMA-U | Variation | Variation (%) |
|---|---|---|---|---|---|
| False alarms | Warning level 1 | 780 | 721 | -59 | -8% |
| | Warning level 2 | 287 | 197 | -90 | -31% |
| | Warning level 3 | 21 | 11 | -10 | -48% |
| Missed alarms | Number of alarms | 88 | 69 | -19 | -22% |
| | Number of missed landslides | 214 | 134 | -80 | -37% |
| Hits | Number of landslides | 709 | 789 | +80 | +11% |
| | % of total landslides | 76.8 | 85.5 | +8.7 | +11% |

