# Peer review of "Brief communication: Using punctual soil moisture estimates to improve the performances of a regional scale landslide early warning system"

_Natural Hazards and Earth System Sciences, 2017_

## Referee Comment (RC1) · Anonymous Referee #1 · 21 Nov 2017

Dear Editor,

Thank you for the opportunity to peer review the paper with title "Using punctual soil moisture estimates to improve the performances of a regional scale landslide early warning system".

In this paper the authors have demonstrated using the mean soil moisture and SIGMA-U approach for improvement of regional scale landslide early warning system in the Emilia Romagna Region (Northern Italy). Authors have attempted to reduce numbers of false and missed alarms by the back analysis using landslide events, soil data and rainfall data from the period of 2011 and 2014. From the content as a whole it can

be seen that described method and procedure can be integrated into the landslide warning system but further tests are needed before.

General referee comment The objectives defined by the authors are quite clear and paper is good structured and the reader can distinguish between material and methods, results and discussion. The drawback in this manuscript is lack of detailed review of literature about the importance of the soil moisture and antecedent rainfall period that significantly influence on triggering landslides. The authors just mentioned the Italian researches and totally overlooked the important researches form the other European and non-European countries where different natural background prevails as well different climate regime (Kim et al., 1992; Heyerdahl et a., 2003; Crozier, 1999; Glade et al., 2000; Aleotti, 2004, Chleborad, 2003, Zezere, 2005, Jemec Auflič and Komac, 2013, etc.). The authors should also improve mean soil moisture values by means of reviewing also rainfall events that not triggered landslides where amount of rainfall was above the rainfall threshold as well indicate why each TU has the same MSM value. According to the above mentioned facts the present paper will be ready for publication after major revisions.

Here are listed specific comments that I would recommend the authors makes.

Page 1 Line 25: Cardinali et al. 2006 is not listed in the chapter of References Page 3 Line 9: Please explain how you know "under which landslides never triggered". Have you done any correlation that for the defined MSM threshold landslides never occurred? Line 14: Please explain and add why you set MSM =75% equal for all TUs? There is no evidence for this. Moreover if the geological setting in each TU is different there must be a difference in MSM values per TUs then.

Linguistic alterations In general the manuscript is written in acceptable English, but some sentences have to be rewritten. Nonetheless, the entire document should be revised by a native speaker.

Kind regards, reviewer

---

## Referee Comment (RC2) · Anonymous Referee #2 · 10 Dec 2017

**General comments**

The paper briefly communicates the improvement of a previous version of a landslide early warning decision tree (SIGMA) by adding soil moisture information. Two separate methodologies are presented. The first consists in cutting-off the application of SIGMA if mean daily soil moisture (MSM) averaged on the given Territorial Unit (TU) is below a threshold value. The second uses the time series of soil moisture measured at a point within the decision tree of SIGMA. The topic fits within the scope of NHESS. The paper is globally well-written, though language is improvable. However, I have some concerns about the real improvement obtained by using soil moisture information, and I think that

the authors should prove the improvement by more in-depth tests. In particular, the authors should address the following points:

- As far as I understand, MSM is available from TOPKAPI for all (or most of) the 25 TUs. Why the authors apply it only to 7 selected TUs? This could be an ad hoc choice to make the methodology work well

- Soil moisture measured at an arbitrary point (where are the punctual measurements located?), may be totally unrelated to soil moisture at landslide locations. Hence the improvement showed by the authors may be just a case. For a more robust testing, the authors should apply some sort of "jack-knife" validation test

Another point is that I do not see the rationale of considering the standard deviation of a random variate as an indication of its magnitude. The standard deviation is a measure of dispersion. The magnitude could be rather expressed by comparing the difference between the value and the mean with the standard deviation.

For the reasons above I think that this brief communication should undergo **major revisions** before its publication.

**Specific comments**

P3 from L18. "A back analysis...". Why only 7 TUs are used for the test?

P3 L19 "from 320 to 231" these numbers differ from those in table 1. That's okay because, as far as I understand, the number of TUs considered is different in the two cases. Maybe the authors should explain better this point

P4 L3: I understand that the SIGMA model has already been published by the authors, but the rationale of using standard deviation is not clear. The authors should possibly explain better this point. (See general comments)

P4 from L14 "The results of the back-analysis clearly show an overall improvement..." The authors should apply a more in-depth test for assessing that the performances

truly improve, by applying a "jack-knife"/"leave one out" validation test. This consists in the following: a) calibrate the decision tree based on all rainfall events except one (left-out); b) test the performance of the calibrated decision tree on the rainfall event left-out; c) repeat steps a) and b) until all rainfall events are covered as left-outs, d) summarize the results (e.g. by ROC indices) of all the left-outs. This may be done for all TUs. Other similar validation tests may be applied (See e.g. Haykin, 1997)

*Haykin, S., 1999. Neural Networks: A Comprehensive Foundation. Prentice Hall, Upper Saddle River, New Jersey.*

P1 L17 Possibly update references on landslide triggering thresholds by adding, e.g.: Peruccacci et al, 2017; Peres and Cancelliere, 2016; Leonarduzzi et al., 2017.

*Leonarduzzi, E., Molnar, P. and Mcardell, B. W.: Predictive performance of rainfall thresholds for shallow landslides in Switzerland from gridded daily data, doi:10.1002/2017WR021044, 2017.*

*Peres, D. J. and Cancelliere, A.: Estimating return period of landslide triggering by Monte Carlo simulation, J. Hydrol., doi:10.1016/j.jhydrol.2016.03.036, 2016.*

*Peruccacci, S., Brunetti, M. T., Gariano, S. L., Melillo, M., Rossi, M. and Guzzetti, F.: Rainfall thresholds for possible landslide occurrence in Italy, Geomorphology, 290, 39–57, doi:10.1016/j.geomorph.2017.03.031, 2017.*

Perhaps the introduction may take into account that the importance of including soil moisture information in landslide triggering thresholds has been stressed by a recent NHESS invited perspective by Bogaard and Greco, 2017.

*Bogaard, T. and Greco, R.: Invited perspectives. A hydrological look to precipitation intensity duration thresholds for landslide initiation: proposing hydro-meteorological thresholds, Nat. Hazards Earth Syst. Sci. Discuss., 1–17,doi:10.5194/nhess-2017-241, 2017.*

Tab. 1: also the number of landslides and true positives and negatives should be

shown, and commented in the text

**Technical corrections**

P1 L12 maybe replace "were" with "are"

P1 L22 "thresholds" instead of "threshold"

P1 L23 "landslide occurrence" instead of "the landslide occurrence"

P1 L25 "as a proxy" instead of "a proxy"

P2 L2 "landslide" instead of "the landslide"

P2 L29 here introduce the acronym MSM

P2 L27 "rainfall-runoff" instead of "inflow-outflow"

P4 L14 "importantly" instead of "important"

P4 L29 "is by large" maybe can be improved

Fig. 1: Where soil punctual measurements were taken?

Fig. 2: on the upper-left: there must be a mistake in the orientation of the arrows

---

## Author Comment (AC2) · 30 Dec 2017

We express our gratitude to the Reviewer, which pointed out some weaknesses of the manuscript and gave us several insights to improve it.
In the following, we provide a point-to-point reply (AA – authors' answers) to every referee comment (RC).

-

General comments

RC – The paper briefly communicates the improvement of a previous version of a landslide early warning decision tree (SIGMA) by adding soil moisture information. Two separate methodologies are presented. The first consists in cutting-off the application of SIGMA if mean daily soil moisture (MSM) averaged on the given Territorial Unit (TU) is below a threshold value. The second uses the time series of soil moisture measured at a point within the decision tree of SIGMA. The topic fits within the scope of NHESS. The paper is globally well-written, though language is improvable. However, I have some concerns about the real improvement obtained by using soil moisture information, and I think that the authors should prove the improvement by more in-depth tests. In particular, the authors should address the following points:

AA – During the revision stage, we will improve the writing and we will address all the points mentioned by the Referee.

-

RC – • As far as I understand, MSM is available from TOPKAPI for all (or most of) the 25 TUs. Why the authors apply it only to 7 selected TUs? This could be an ad hoc choice to make the methodology work well.

AA – A few words to explain the difference in the test sites between the two experiments (MSM experiment and Sigma-U experiment): during the first stage of the research we had at our disposal only soil moisture data from 7 TUs (years 2009-2014). There, we tested the MSM approach. Results were deemed encouraging, therefore when we obtained an increased dataset of soil moisture data (7 more TUs, but limited to the years 2011-2014) we directly developed and tested a more elaborate approach (the Sigma-U experiment).
Since in this work we are describing two distinct experiments, we decided to use two distinct datasets, related to test sites of different extension.
Concerning this issue, we ask the suggestion also from the Handling Editor: we have the raw data and we could extend the back-analysis also to the additional 7 TUs, but in there, the timespan would be limited to the years 2011-2014.

-

RC – • Soil moisture measured at an arbitrary point (where are the punctual measurements located?), may be totally unrelated to soil moisture at landslide locations. Hence the improvement showed by the authors may be just a case. For a

more robust testing, the authors should apply some sort of "jack-knife" validation test.

AA – Unfortunately, a misunderstanding occurred. We do not use measurements. In the manuscript, we were very careful to use the term "punctual estimates", as values are not actually measured (e.g. by instruments or monitoring stations): they are estimated by TOPKAPI model. We use "punctual" to stress that we are not performing a distributed assessment [e.g. on a pixel-by-pixel basis]: since the original EWS uses only a rainfall measuring station for each territorial unit, we need only a soil moisture value for each territorial unit. That's a value averaged for the whole TU, consistently with the "reference rain gauge" approach in which a rain gauge provides a rainfall value considered representative for a whole territorial unit. This could be clarified in the "materials and method" section, which could be edited as follows:
*"For most of the hydrographic basins of the region, ARPAE-ER (…) provides the mean soil moisture value at hourly time step. These are values estimated by the TOPKAPI (…) model (…), which is a rainfall-runoff model that can provide high resolution hydrological information. We use these data to estimate the mean daily soil moisture (MSM) value for each TU…"*

Please, consider also that we need to use only data readily available online to be used in real time in the EWS. ARPAE-ER does not provided distributed soil moisture data, it provides aggregated soil moisture data and they are just what we need for our objectives.

-

RC – Another point is that I do not see the rationale of considering the standard deviation of a random variate as an indication of its magnitude. The standard deviation is a measure of dispersion. The magnitude could be rather expressed by comparing the difference between the value and the mean with the standard deviation.

AA – Maybe we over-simplified the description of the original model SIGMA. In the revised version of the manuscript, more space will be devoted to the description of SIGMA approach and the passage from daily time series to sigma curves.

-

RC – For the reasons above I think that this brief communication should undergo major revisions before its publication.

AA – We thank the referee for the constructive comments, we will work hard on the revised version of the manuscript.

-

Specific comments

RC – P3 from L18. "A back analysis. . .". Why only 7 TUs are used for the test?

AA – As we explained in response to a previous comment, the MSM experiment was performed on the first dataset we had at our disposal: 7 TUs, years 2009-2014. The SigmaU experiment was performed on 7+7 TUs, years 2011-2014.

-

RC – P3 L19 "from 320 to 231" these numbers differ from those in table 1. That's okay because, as far as I understand, the number of TUs considered is different in the two cases. Maybe the authors should explain better this point.

AA – The main difference is not the TU number, it is that we are making comparisons between different approaches. The text highlighted in the referee comment (P3 L19) is placed in section 3.1 and it is about the difference between Sigma and the MSM approach. Table 1 is referred to section 3.2 and it is about the difference between Sigma and the Sigma-U approach. Since we are comparing Sigma with two different approaches, it is normal that numbers are different. We believe that the misunderstanding arose because figures and tables are listed at the end of the manuscript. In the final edited paper the table will be placed at the right point in the text and we think that will be sufficient to avoid misunderstandings.

-

RC – P4 L3: I understand that the SIGMA model has already been published by the authors, but the rationale of using standard deviation is not clear. The authors should possibly explain better this point. (See general comments).

AA – Maybe we over-simplified the description of the original model. In the revised version of the manuscript, more space will be devoted to the description of the SIGMA approach and the passage from daily time series to sigma curves.

-

RC – P4 from L14 "The results of the back-analysis clearly show an overall improvement. . ." The authors should apply a more in-depth test for assessing that the performances truly improve, by applying a "jack-knife"/"leave one out" validation test. This consists in the following: a) calibrate the decision tree based on all rainfall events except one (left-out); b) test the performance of the calibrated decision tree on the rainfall event left-out; c) repeat steps a) and b) until all rainfall events are covered as left-outs, d) summarize the results (e.g. by ROC indices) of all the left-outs. This may be done for all TUs. Other similar validation tests may be applied (See e.g. Haykin, 1997).
Haykin, S., 1999. Neural Networks: A Comprehensive Foundation. Prentice Hall, Upper Saddle River, New Jersey.

AA – In this manuscript, we use a different method, more simple and more straightforward than suggested by the Referee, but still we believe it could provide a rigorous quantitative assessment of the performances/improvements. We formulate a hypothesis (EWS can be enhanced by substituting antecedent rainfall with soil moisture) and we empirically verify that it is met in our testing dataset. We also shows basic statistics (count of hits and errors). As we stated in the conclusion, before actual implementation in the EWS, additional data should be gathered for a more robust calibration should, possibly including one ad-hoc threshold for each TU (and not the same threshold for the whole region). At that time, the approach suggested by the Referee will provide a valuable contribution. However, we agree with the reviewer that the sentence "*The results of the back-analysis clearly show an overall improvement*" is too "definitive" and would need more robust support. Therefore, we believe that the text should be modified with something like "*The results of the back-analysis are encouraging, as the count of both false alarms and missed alarms is lower in SIGMA-U than in SIGMA*".

-

RC – P1 L17 Possibly update references on landslide triggering thresholds by adding, e.g.: Peruccacci et al, 2017; Peres and Cancelliere, 2016; Leonarduzzi et al., 2017.
Leonarduzzi, E., Molnar, P. and Mcardell, B. W.: Predictive performance of rainfall thresholds for shallow landslides in Switzerland from gridded daily data, doi:10.1002/2017WR021044, 2017.
Peres, D. J. and Cancelliere, A.: Estimating return period of landslide triggering by Monte Carlo simulation, J. Hydrol., doi:10.1016/j.jhydrol.2016.03.036, 2016.
Peruccacci, S., Brunetti, M. T., Gariano, S. L., Melillo, M., Rossi, M. and Guzzetti, F.: Rainfall thresholds for possible landslide occurrence in Italy, Geomorphology, 290, 39–57, doi:10.1016/j.geomorph.2017.03.031, 2017.

AA – Thank you for the suggestion, we will perform an update to the references cited in the manuscript. Please note that the "rainfall thresholds" topic is very broad, therefore we focused the introduction on literature thresholds that are implemented into EWS and on thresholds considering antecedent rainfall as a proxy for soil moisture conditions. We were forced to a very strict focus because the manuscript typology demand a limitation to max. 20 references. We will carefully go through the suggested papers and include them in the introduction, when appropriate.

-

RC – Perhaps the introduction may take into account that the importance of including soil moisture information in landslide triggering thresholds has been stressed by a recent NHESS invited perspective by Bogaard and Greco, 2017.
Bogaard, T. and Greco, R.: Invited perspectives. A hydrological look to precipitation intensity duration thresholds for landslide initiation: proposing hydro-meteorological thresholds, Nat. Hazards Earth Syst. Sci. Discuss., 1–17,doi:10.5194/nhess-2017-

241, 2017.

AA – Indeed, the Referee suggests a very interesting article. We will make reference to it in the introduction and also in the conclusion, since we believe that our works expands by a small step the classical rainfall threshold approach towards the direction expressed by Bogaard and Greco: instead of using only rainfall, we try to better encompass the hydrology of the territorial units by using soil moisture. On this regard, we will also make reference to a work very recently submitted to the same special issue by Kanli et al. (2017), which shares a similar perspective.
Canli, E., Mergili, M., and Glade, T.: Probabilistic landslide ensemble prediction systems: Lessons to be learned from hydrology, Nat. Hazards Earth Syst. Sci. Discuss., https://doi.org/10.5194/nhess-2017-427, in review, 2017.
We were not aware of these papers when we submitted the first version of our manuscript.

-

RC – Tab. 1: also the number of landslides and true positives and negatives should be shown, and commented in the text

AA – In the revised version of the manuscript, we will expand the table as suggested and provide the necessary comments in the text.
While performing calculations about true positives, we noticed an error: the total number of landslides (hits+missed) was not the same in SIGMA and SIGMA-U. After a thorough check of the used spreadsheet, we identified an error in the formulas: in a few words, 5 TUs were erroneously not included in the calculations for Table 1. We corrected the formulas and the links and re-calculated the statistics, which now result even better than the mistaken ones reported in the previous version of the manuscript:
"…*false alarms issued at warning level 1, which are negligible, decreased by 8%, while the very important warning level 3 was erroneously issued 11 times instead of 21 (-48%). False alarms at the intermediate warning level 2 were reduced from 287 to 197 (-31%). Missed alarms are reduced as well: while SIGMA missed 88 alarms, SIGMA-U missed 69 alarms (-22%). This corresponds to a total of 134 missed landslides instead of 214 (-37%). Overall, SIGMA-U hits 789 landslides out of 923 (85.5%), outperforming SIGMA that hits 709 landslides (76.8%)."*
We apologize for the error and we express our gratitude to the Referee that made it possible to notice it and to correct it.

-

RC – Technical corrections
P1 L12 maybe replace "were" with "are"
P1 L22 "thresholds" instead of "threshold"
P1 L23 "landslide occurrence" instead of "the landslide occurrence"
P1 L25 "as a proxy" instead of "a proxy"
P2 L2 "landslide" instead of "the landslide"

P2 L29 here introduce the acronym MSM
P2 L27 "rainfall-runoff" instead of "inflow-outflow"
P4 L14 "importantly" instead of "important"
P4 L29 "is by large" maybe can be improved

AA – All suggested corrections have been included in the revised text.

-

RC – Fig. 1: Where soil punctual measurements were taken?

AA – As explained before, we do not use measurements. In the manuscript, we use the term "punctual estimates". Values are not actually measured, they are estimated by TOPKAPI model. We use "punctual" to stress that we are not performing a distributed assessment [e.g. on a pixel-by-pixel basis]: since the original EWS uses only a rainfall measuring station for each territorial unit, we need only a soil moisture value for each territorial unit.
In the conclusions section, we only hypothesize the possibility of using actual measures in the future developments of the research (of course, provided the funds are renewed and the research plan is approved).
Maybe in the introduction a sentence could be misleading *("This work explores the possibility to exploit punctual soil moisture values estimated at few discrete points and to correlate them with landslide triggering over wide areas")*. We could change it to avoid misunderstandings *("This work explores the possibility to exploit punctual soil moisture estimates and to correlate them with landslide triggering over wide areas")*. Maybe, the term "*aggregated*" could be used instead of "*punctual*", if Editor and Referees consider it clearer.

-

RC – Fig. 2: on the upper-left: there must be a mistake in the orientation of the arrows

AA – Thank you for identifying this issue. The figure was adjusted.

-

On behalf of all authors,
Samuele Segoni

---

## Author Response (AR1)

**Manuscript nhess-2017-361, first submitted on 11 Oct 2017**

**(Brief communication: Using punctual soil moisture estimates to improve the performances of a regional scale landslide early warning system)**

Dear Editor and Referees, thank you for your comments and insights that in my opinion contributed to improve the manuscript.

You can find in this document a point-to point reply to every referee comment (including modifications performed to the text) and a track-change version of the manuscript.

Kind regards,

Samuele Segoni (on behalf of all co-authors)

ANSWERS TO REFEREE #1

We express our gratitude to the Reviewer, which pointed out some weaknesses of the manuscript and gave us insights to improve it.

In the following text, we provide a point-to-point reply (AA – authors' answers) to every referee comment (RC).

-

RC - In this paper the authors have demonstrated using the mean soil moisture and SIGMAU approach for improvement of regional scale landslide early warning system in the Emilia Romagna Region (Northern Italy). Authors have attempted to reduce numbers of false and missed alarms by the back analysis using landslide events, soil data and rainfall data from the period of 2011 and 2014. From the content as a whole it can be seen that described method and procedure can be integrated into the landslide warning system but further tests are needed before.

AA - The referee centered the point: we performed a back analysis to reduce false and missed alarms by integrating soil moisture measures into a warning system based on rainfall thresholds. We believe that our work proves, with the evidence of data, that the approach is feasible and a reduction of alarms can be obtained. This outcome represents an important intermediate step in our research activity, this is why we selected the "short communication" manuscript type when submitting our work.

-

RC - The objectives defined by the authors are quite clear and paper is good structured and the reader can distinguish between material and methods, results and discussion.

AA – Thank you for appreciating the structure of the manuscript.

-

RC – The drawback in this manuscript is lack of detailed review of literature about the importance of the soil moisture and antecedent rainfall period that significantly influence on triggering landslides. The authors just mentioned the Italian researches and totally overlooked the important researches form the other European and non-European countries where different natural background prevails as well different climate regime (Kim et al., 1992; Heyerdahl et a., 2003; Crozier, 1999; Glade et al., 2000; Aleotti, 2004, Chleborad, 2003, Zezere, 2005, Jemec Auflič and Komac, 2013, etc.).

AA – We agree with the Referee and we are aware of this drawback. Unfortunately, the manuscript typology (short communication) gave us some limitations (text length, number of references) and we decided to focus the introduction on a limited number of works, with a background similar to our case of study (regional scale analysis, application to EWS). We agree with Referee's comment and in the revised version of the manuscript, we fully addressed this comment, providing an extended

literature review with insights on almost all the suggested references (the ones published in international journals). In addition, some Italian references were deleted to devote more space to works from other parts of the world and to limit the total number of references as requested in the "short communication" manuscript typology.

Please, let us know if you believe we left out some relevant references about thresholds using antecedent rainfall or thresholds integrated into EWS.

**PREVIOUS VERSION OF THE TEXT**

Regional scale landslide early warning systems (RSLEWS henceforth) are usually based on empirical rainfall threshold, which in turn are based on rainfall parameters easy to measure and monitor by means of rain gauges (Guzzetti et al., 2007; Baum et al., 2010; Cannon et al., 2011; Segoni et al., 2015a; Piciullo et al., 2017).

However, it is widely recognized that soil moisture conditions before the triggering rainfall event can play a crucial role in the initiation of landslides, especially for deep-seated landslides and for terrains with complex hydrological settings (Wieczorek, 1996; Martelloni et al., 2012).

Unfortunately, the influence of soil moisture conditions is difficult to be adequately considered in RSLEWS. One of the most widespread approaches is establishing rainfall threshold based on the rainfall amount accumulated during a given period before the landslide occurrence or before the triggering rainfall event (Guzzetti et al., 2007, and references therein). The length of these timespans varies widely in the international literature, e.g. from a few days (Calvello et al., 2015) to a few months (Cardinali et al. 2006).

**REVISED TEXT:**

Regional scale landslide early warning systems (RSLEWS henceforth) are usually based on empirical rainfall thresholds, which in turn are based on rainfall parameters that can be easily measured and monitored by rain gauges (Aleotti, 2004; Baum et al., 2010; Cannon et al., 2011; Segoni et al., 2015a; Leonarduzzi et al., 2017; Piciullo et al., 2017).

However, it is widely recognized that soil moisture conditions before the triggering rainfall event can play a crucial role in the initiation of landslides, especially if deep-seated landslides and terrains with complex hydrological settings are involved (Wieczorek, 1996; Zezere et al., 2005; Jemec and Komac, 2013; Peres and Cancelliere, 2016; Bogaard and Greco, 2017).

Unfortunately, the influence of soil moisture conditions is difficult to be encompassed into RSLEWS. One of the most widespread approaches is establishing rainfall thresholds based on the rainfall amount accumulated during a given period before landslide occurrence or before the triggering rainfall event (Kim et al., 1991; Chleborad, 2003). The length of these timespans varies widely in the international literature, e.g. from a few days (Kim et al., 1991; Calvello et al., 2015) to a few months (Zezere et al., 2005). More advanced models combine daily rainfall data to compute antecedent rainfall indexes that can be used to forecast landslide occurrence (Cozier, 1999; Glade et al., 2000).

Please, also refer to the track-changes document for other modifications to the references used.

-

RC – The authors should also improve mean soil moisture values by means of

reviewing also rainfall events that not triggered landslides where amount of rainfall was above the rainfall threshold (..)

AA – What the referee calls "rainfall events that not triggered landslides where amount of rainfall was above the rainfall threshold", is reported in the text as "false alarms". They are fully considered in the test performed by means of the back analysis. Maybe a misunderstanding has arisen because we didn't define missed alarms and false alarms in the previous version of the text. Now we have modified the text accordingly. In addition, please note that the use of MSM threshold described in 3.1 would never be capable of reducing the false alarms committed by SIGMA, as it acts like a cut-off. In a few words, it reduces the alarms issued by SIGMA, but it does not allow SIGMA to issue additional alarms.

PREVIOUS VERSION OF THE TEXT

*A back analysis performed for the years 2009-2014 over the 7 test TUs shows a marked reduction of false alarms: false alarms in the first warning level decrease from 320 to 231 (-28%), false alarms in the second warning level decreases from 169 to 141 (-17%) and false alarms in the third warning level decreases from 13 to 5 (-62%). To correctly evaluate the effectiveness of a EWS, the improvement concerning false alarms should be weighed against the behavior concerning missed alarms. We verified that the introduction of the MSM threshold causes the increase of false alarm counts only by 1. The already mentioned event occurred in 01/06/2013, consisting in three landslides (lowest alarm level according to Lagomarsino et al., 2013). Since this was a very minor event and since lowering the MSM threshold to 54% would result in an almost total loss of the benefits in terms of false alarm reduction, the 75% threshold was considered successfully tested and the 01/06/2013 event was considered an acceptable tradeoff in the light of a general improvement of the warning system.*

MODIFIED VERSION OF THE TEXT

A back analysis performed for the years 2009-2014 over the 7 test TUs shows a marked reduction of false alarms (days in which the rainfall thresholds are exceeded but no landslides are reported). More in detail: false alarms in the first warning level decreased from 320 to 231 (-28%), false alarms in the second warning level decreased from 169 to 141 (-17%) and false alarms in the third warning level decreased from 13 to 5 (-62%). To correctly evaluate the effectiveness of a EWS, the improvement concerning false alarms should be weighed against the behavior concerning missed alarms (days in which the rainfall thresholds are not exceeded but landslides are reported). We verified that the introduction of the MSM threshold caused the increase of missed alarm counts only by 1: the already mentioned event occurred in 01/06/2013, consisting in three landslides (lowest alarm level according to Lagomarsino et al., 2013). Since this was a minor event and since lowering the MSM threshold to 54% would result in an almost total loss of the benefits in terms of false alarm reduction, the 75% threshold was considered successfully tested and the 01/06/2013 event was considered an acceptable trade-off for a general improvement of the warning system.

It should be noted that the described use of the MSM threshold is not capable of reducing the missed alarms committed by SIGMA, as it acts like a cut-off filter. To obtain a reduction of both missed and false alarms, a more radical modification of SIGMA is depicted in the next section.

-
RC – (…) as well indicate why each TU has the same MSM value.

AA – Please note that in every moment, MSM is different for each TU. What is equal in each TU is the MSM value used as a threshold in the EWS. When we took this decision, we had two options: (1) a MSM threshold value different for each TU; (2) same MSM threshold value in each TU. In an optimal condition, we agree with the referee that the first option would be preferable. However, a threshold value requires experimental data (i.e. landslide events) to be considered robust. We had the problem of several TUs with only few landslide events. For example, TU21 has only 4 landslide events. A purposely developed threshold would be characterized by a very weak empirical correlation. In our opinion, a threshold calibrated against only 4 events cannot be considered valid and cannot be safely used in an operational warning system. We therefore decided to renounce at the "detail" of the personalized threshold in favor of a more robust threshold generalized for the whole test area. The value used as threshold (75%) is not selected arbitrarily, but it is the mean of the values encountered in each TU. Please also note that the tests performed on the back analysis highlighted that our choice reached the objective of reducing false alarms. We modified the text to consider this issue and to address the Referee's comment.

PREVIOUS VERSION OF THE TEXT
*We decided to modify SIGMA algorithm using a threshold based on MSM = 75%, equal for all TUs. Basically, the modified version of the algorithm checks the MSM value and uses the module of rainfall only if MSM>75%. Under this threshold, no landslide is expected and the original SIGMA algorithm based on rainfall thresholds does not starts. Above the threshold, landslides could be expected if particular rainfall conditions are verified, therefore SIGMA algorithm is launched.*

MODIFIED VERSION OF THE TEXT
*We modified SIGMA algorithm adding a cut-off threshold defined as MSM = 75%, which is the arithmetic mean of the values of each TU. Basically, the modified version of the algorithm checks the daily MSM value reported for a given TU, and compares it with the MSM=75% threshold. Under this value, no landslide is expected and the SIGMA algorithm is not launched. If daily MSM is higher than 75%, landslides could be expected if particular rainfall conditions are verified, therefore SIGMA algorithm is launched. We set a MSM threshold equal for all TUs because in some TUs the landslide dataset contains only a few events (e.g. only 4 landslide events in TU21) and a dedicated MSM threshold value would be characterized by a very weak empirical correlation that would prevent a safe use in the RSLEWS. In addition, if we exclude the outliers, all TUs are characterized by small variations in MSM threshold values (from 75% to 79%). We therefore decided to renounce at the "detail" of the personalized threshold in favor of a more robust MSM threshold generalized for the whole test area.*

-
RC – According to the above mentioned facts the present paper will be ready for publication after major revisions.
Here are listed specific comments that I would recommend the authors makes.

AA – We deeply modified the text, addressing all issues reported in the previous general comments and in the specific comments hereafter. As a result, sections 1 and 3.1 have been deeply modified. Also sections 2, 3.2 and 4 and have been modified according to the suggestions of Referee#2. All amendments will be highlighted in the revised text.

-
RC – Page 1 Line 25: Cardinali et al. 2006 is not listed in the chapter of References

AA – Since we were criticized to have used too many Italian references, and since we needed to reduce the reference number, this reference was removed.

-
RC – Page 3 Line 9: Please explain how you know "under which landslides never triggered". Have you done any correlation that for the defined MSM threshold landslides never occurred?

AA – Yes, we did a correlation and we empirically verified what we stated. The revised text is more clear on this point, providing a more in-depth description and showing data. Of course, we refer to the landslides reported in the dataset.
PREVIOUS VERSION OF THE TEXT:
We compared all landslide occurrences in the years 2009-2014 and MSM (mean soil moisture) at each TU. We verified that for each TU a threshold MSM value can be identified under which landslides are never triggered, independently from the rainfall amount.

MODIFIED VERSION OF THE TEXT:
*We compared all landslide occurrences in the years 2009-2014 and MSM (mean soil moisture) at each TU. We verified that for each TU a threshold MSM value can be identified under which landslides have never been reported, independently from the rainfall amount. In addition, we verified that in general TUs had similar threshold MSM, with a few exceptions. Threshold MSM is 75% in TU23 and TU22, 76% in TU18, 78% in TU17, and 79% in TU19. In TU21, the threshold MSM is 88%. This value is higher than all other TUs and it can be partially explained with the scarcity of data: only 4 landslide events are included in the testing dataset of TU21. TU20 presents a landslide event with 54% MSM. If we consider this event as an outlier and we exclude it from the analysis, the value is 75% also for TU20.*

-
RC – Line 14: Please explain and add why you set MSM =75% equal for all TUs? There is no evidence for this. Moreover if the geological setting in each TU is different there must be a difference in MSM values per TUs then.

AA – Actually, we found enough evidence but we acknowledge that we did not show it adequately in the previous version of the text. Now we deeply modified the text, showing data and enhancing the description.
In brief, there are three reasons why we set 75% for all TUs:
-   It is the mean value (now data are shown with greater detail and this point can be easily verified)
-   In almost all TUs MSM thresholds are very similar (75%-79%)
-   Significantly higher MSM can be found in TU21, but taking this value as a threshold is not feasible because it would be calibrated against a very scarce test sample (see also answer to general comment on this issue).
About the question "why 75%?", the Referee is absolutely right: why choosing a 75% threshold if a lower value (73%) is found? This comment allowed us to identify a typo in the text: In the sentence "The MSM threshold varies generally from 73% (TU 23) to 88% (TU 21)", the number 73 was wrong (probably just a typo): the

correct value is "75% (TU 23)". That explains why we used the 75% value: because it was the lowest threshold found in our test dataset (of course excluding the outlier) and because it represents the mean value (considering the outlier 54% and the 88% value influenced by scarcity of data). Now the old sentence is not part of the text anymore because we deeply modified the section: all MSM threshold values are listed and it could be seen that 75% is the mean MSM threshold.

In addition, stimulated by this comment, we searched for a correlation between MSM threshold values and environmental characteristics of the TUs (average slope, average and prevailing aspect class, lithology). We didn't find a clear correlation, maybe just because the MSM threshold range is very narrow (75-79%). This outcome strengthened our belief that the 88% outlier is not due to environmental characteristics but to the scarcity of data. Hence, one more reason to adopt a single threshold value for the whole test area.

PREVIOUS VERSION OF THE TEXT:
*As a consequence, taking this limit into account could prevent SIGMA from committing false alarms in case of abundant rainfalls outside the rainy season, when the soil is dry. The MSM threshold varies generally from 73% (TU 23) to 88% (TU 21). The only exception to this rule is TU 20, where an event of 3 landslides occurred in 01/06/2013 with a MSM of 54%, although all the other landslides of the TU occurred with MSM equal or higher than 75%.*

*We decided to modify SIGMA algorithm using a threshold based on MSM = 75%, equal for all TUs. Basically, the modified version of the algorithm checks the MSM value and uses the module of rainfall only if MSM>75%. Under this threshold, no landslide is expected and the original SIGMA algorithm based on rainfall thresholds does not starts. Above the threshold, landslides could be expected if particular rainfall conditions are verified, therefore SIGMA algorithm is launched.*

MODIFIED VERSION OF THE TEXT:
*We verified that for each TU a threshold MSM value can be identified under which landslides have never been reported, independently from the rainfall amount. In addition, we verified that in general TUs had similar threshold MSM, with a few exceptions. Threshold MSM is 75% in TU23 and TU22, 76% in TU18, 78% in TU17, and 79% in TU19. In TU21, the threshold MSM is 88%. This value is higher than all other TUs and it can be partially explained with the scarcity of data: only 4 landslide events are included in the testing dataset of TU21. TU20 presents a landslide event with 54% MSM. If we consider this event as an outlier and we exclude it from the analysis, the value is 75% also for TU20.*
*Consequently, taking a MSM threshold into account could prevent SIGMA from committing false alarms in case of abundant rainfalls outside the rainy season, when the soil is dry. Therefore, we modified SIGMA algorithm adding a cut-off threshold defined as MSM = 75%, which is the arithmetic mean of the values of each TU. Basically, the modified version of the algorithm checks the daily MSM value reported for a given TU, and compares it with the MSM=75% threshold. Under this value, no landslide is expected and the SIGMA algorithm is not launched. If daily MSM is higher than 75%, landslides could be expected if particular rainfall conditions are verified, therefore SIGMA algorithm is launched. We set a MSM threshold equal for all TUs because in some TUs the landslide dataset contains only a few events (e.g. only 4 landslide events in TU21) and a dedicated MSM threshold value would be characterized by a very weak empirical correlation that would prevent a safe use in the RSLEWS. In addition, if we exclude the outliers, all TUs are characterized by small variations in MSM threshold values (from 75% to 79%). We therefore decided to renounce at the "detail" of the personalized threshold in favor of a more robust MSM threshold generalized for the whole test area.*
-

RC – Linguistic alterations In general the manuscript is written in acceptable English, but some sentences have to be rewritten. Nonetheless, the entire document should be revised by a native speaker.

AA – The text was revised by an expert. She performed minor corrections, changed some terms and adjusted some awkward sentences.

ANSWERS TO REFEREE #2

We express our gratitude to the Reviewer, which pointed out some weaknesses of the manuscript and gave us several insights to improve it.
In the following, we provide a point-to-point reply (AA – authors' answers) to every referee comment (RC).

-
General comments

RC – The paper briefly communicates the improvement of a previous version of a landslide early warning decision tree (SIGMA) by adding soil moisture information. Two separate methodologies are presented. The first consists in cutting-off the application of SIGMA if mean daily soil moisture (MSM) averaged on the given Territorial Unit (TU) is below a threshold value. The second uses the time series of soil moisture measured at a point within the decision tree of SIGMA. The topic fits within the scope of NHESS. The paper is globally well-written, though language is improvable. However, I have some concerns about the real improvement obtained by using soil moisture information, and I think that the authors should prove the improvement by more in-depth tests. In particular, the authors should address the following points:

AA –The text was revised by an expert. She performed minor corrections, changed some terms and adjusted some awkward sentences. In the revised version of the manuscript, we addressed all the points mentioned by the Referee.

-

RC – • As far as I understand, MSM is available from TOPKAPI for all (or most of) the 25 TUs. Why the authors apply it only to 7 selected TUs? This could be an ad hoc choice to make the methodology work well.

AA – A few words to explain the difference in the test sites between the two experiments (MSM experiment and Sigma-U experiment): during the first stage of the research we had at our disposal only soil moisture data from 7 TUs (years 2009-2014). There, we tested the MSM approach. Results were deemed encouraging, therefore when we obtained an increased dataset of soil moisture data (7 more TUs, but limited to the years 2011-2014) we directly developed and tested a more elaborate approach (the Sigma-U experiment). MSM approach should be considered a preliminary test.
Since in this work we are describing two distinct experiments, we decided to use two distinct datasets, related to test sites of different extension.

-

RC – • Soil moisture measured at an arbitrary point (where are the punctual measurements located?), may be totally unrelated to soil moisture at landslide

locations. Hence the improvement showed by the authors may be just a case. For a more robust testing, the authors should apply some sort of "jack-knife" validation test.

AA – Unfortunately, a misunderstanding occurred. We do not use measurements. In the manuscript, we were very careful to use the term "punctual estimates", as values are not actually measured (e.g. by instruments or monitoring stations): they are estimated by TOPKAPI model. We use "punctual" to stress that we are not performing a distributed assessment [e.g. on a pixel-by-pixel basis]: since the original EWS uses only a rainfall measuring station for each territorial unit, we need only a soil moisture value for each territorial unit. That's a value averaged for the whole TU, consistently with the "reference rain gauge" approach in which a rain gauge provides a rainfall value considered representative for a whole territorial unit. This was clarified in the "materials and method" section, which was edited as follows:

*ORIGINAL TEXT*
*For some of the hydrographic basins of the region, ARPAE-ER (Regional Agency for Prevention, Environment and Energy of Emilia Romagna) provides the mean soil moisture value at hourly time step. These values are estimated by TOPKAPI (TOPographic Kinematic APproximation and Integration) (Ciarapica and Todini, 2002), which is a rainfall-runoff model providing high-resolution hydrological information. We use these data to calculate the mean daily soil moisture value for each TU.*

*REVISED TEXT*
*For some of the hydrographic basins of the region, ARPAE-ER (Regional Agency for Prevention, Environment and Energy of Emilia Romagna) provides the mean soil moisture value at hourly time step. These values are estimated by the TOPKAPI (TOPographic Kinematic APproximation and Integration) model (Ciarapica and Todini, 2002), which is a rainfall-runoff model providing high resolution hydrological information. We used these data to estimate the mean daily soil moisture (MSM) value for each TU.*

Please, consider also that we need to use only data readily available online to be used in real time in the EWS. ARPAE-ER does not provided distributed soil moisture data, it provides aggregated soil moisture data and they are just what we need for our objectives.

In addition, please refer also to the answer to one of the subsequent specific comments, where we describe some adjustments to the text.

Concerning the leave-one-out test. We don't believe it is the best test to perform in this stage of a research of this kind, however we followed the referee's suggestion and tried the test.

| TU | MSM | Leave-it-out mean | Impact in model performance |
|---|---|---|---|
| TU 23 | 0.75 | 75% | none |
| TU 22 | 0.75 | 75% | none |
| TU 18 | 0.76 | 75% | none |
| TU 17 | 0.78 | 75% | none |
| TU 19 | 0.79 | 74% | negligible |
| TU 21 | 0.88 | 73% | negligible |
| TU 20 | 0.54 | 79% | high |
| MEAN | 0.75 | | |

The outcome in our opinion corroborates the choice of using the 75% value:
- It is the mean value (now data are shown with greater detail in the text and this point can be easily verified)
- In almost all TUs MSM thresholds are very similar (75%-79%) and 75% represent the lower bound threshold if the 54% outlier, pertaining to a single event, is excluded.
- A significantly higher MSM can be found in TU21, but taking this value as a threshold is not feasible because it is clearly influenced by the scarcity of data characterizing this TU.

We decided to avoid to include the leave-one-out test in the manuscript, because it needs to be shorter than 6 pages. However, we deeply modified the text of section 3.1 (according also to Referee1 suggestions) to make more clear why and how the MSM threshold was defined.

-

RC – Another point is that I do not see the rationale of considering the standard deviation of a random variate as an indication of its magnitude. The standard deviation is a measure of dispersion. The magnitude could be rather expressed by comparing the difference between the value and the mean with the standard deviation.

AA – Maybe we over-simplified the description of the original model SIGMA. In the revised version of the manuscript, more space has been devoted to the description of SIGMA approach and the passage from daily time series to sigma curves.
Please note that in the Sigma model, standard deviation is not used as a magnitude indicator, but it represents the probability of occurrence of a certain rainfall event (original rainfall data distribution is transformed to a Gaussian distribution).
We modified the text as follow:

PREVIOUS VERSION OF THE TEXT

*One of the instruments used to manage landslide hazard is a RSLEWS called SIGMA, which is based on a complex decisional algorithm considering the overcoming of statistical rainfall thresholds (Martelloni et al.*

*2012). The thresholds are defined in terms of standard deviation (σ) from the mean rainfall amount accumulated during progressively increasing time steps. The algorithm considers two different periods of cumulative rainfall: …*

REVISED VESION OF THE TEXT

*The methodology to develop sigma model (fully described in Martelloni et al. 2012) is based on the hypothesis that anomalous or extreme values of rainfall are responsible for landslide triggering and multiples of the standard deviation (σ) are used as thresholds to discriminate between ordinary and extraordinary rainfall events. To obtain probability values of not exceeding a given rainfall threshold, rainfall time series longer than 50 years are taken into account for each rain gauge. Data of the original rainfall distributions are adapted to a target function chosen as a model (Gaussian distribution in this case). After this conversion, it is possible to define any probability of not overcoming by using standard deviation values, which in turn can be related to the corresponding rainfall value of the original data series.*

*SIGMA algorithm considers two different periods of cumulative rainfall: …*

-

RC – For the reasons above I think that this brief communication should undergo major revisions before its publication.

AA – We thank the referee for the constructive comments, we hope he/she could appreciate the revised version of the manuscript.

-

Specific comments

RC – P3 from L18. "A back analysis. . .". Why only 7 TUs are used for the test?

AA – As we explained in response to a previous comment, the MSM experiment was performed on the first dataset we had at our disposal: 7 TUs, years 2009-2014. The SigmaU experiment was performed on 7+7 TUs, years 2011-2014.

-

RC – P3 L19 "from 320 to 231" these numbers differ from those in table 1. That's okay because, as far as I understand, the number of TUs considered is different in the two cases. Maybe the authors should explain better this point.

AA – The main difference is not the TU number, it is that we are making comparisons between different approaches. The text highlighted in the referee comment (P3 L19) is placed in section 3.1 and it is about the difference between Sigma and the MSM approach. Table 1 is referred to section 3.2 and it is about the difference between Sigma and the Sigma-U approach. Since we are comparing Sigma with two different approaches, it is normal that numbers are different. We

believe that the misunderstanding arose because figures and tables are listed at the end of the manuscript. In the final edited paper, the table will be placed at the right point in the text and we think that it will be sufficient to avoid any misunderstandings.

-

RC – P4 L3: I understand that the SIGMA model has already been published by the authors, but the rationale of using standard deviation is not clear. The authors should possibly explain better this point. (See general comments).

AA – Please, see the response to a similar general comment.

-

RC – P4 from L14 "The results of the back-analysis clearly show an overall improvement. . ." The authors should apply a more in-depth test for assessing that the performances truly improve, by applying a "jack-knife"/"leave one out" validation test. This consists in the following: a) calibrate the decision tree based on all rainfall events except one (left-out); b) test the performance of the calibrated decision tree on the rainfall event left-out; c) repeat steps a) and b) until all rainfall events are covered as left-outs, d) summarize the results (e.g. by ROC indices) of all the left-outs. This may be done for all TUs. Other similar validation tests may be applied (See e.g. Haykin, 1997).
Haykin, S., 1999. Neural Networks: A Comprehensive Foundation. Prentice Hall, Upper Saddle River, New Jersey.

AA – In this manuscript, we use a different method, more simple and more straightforward than suggested by the Referee, but still we believe it could provide a rigorous quantitative assessment of the performances/improvements. We formulate a hypothesis (EWS can be enhanced by substituting antecedent rainfall with soil moisture) and we empirically verify that it is met in our testing dataset. We also show basic statistics (count of hits and errors). As we stated in the conclusion, before actual implementation in the EWS, additional data should be gathered for a more robust calibration, possibly including one ad-hoc threshold for each TU (and not the same threshold for the whole region). At that time, the approach suggested by the Referee will provide a valuable contribution.
However, we agree with the reviewer that the sentence
"*The results of the back-analysis clearly show an overall improvement*"
is too "definitive" and would need more robust support. Therefore, we modified the text with
"*The results of the back-analysis are encouraging, as the count of both false alarms and missed alarms is lower in SIGMA-U than in SIGMA*".
This sentence is not "absolute" as the previous one and it is supported by data.

-

RC – P1 L17 Possibly update references on landslide triggering thresholds by adding, e.g.: Peruccacci et al, 2017; Peres and Cancelliere, 2016; Leonarduzzi et al., 2017.
Leonarduzzi, E., Molnar, P. and Mcardell, B. W.: Predictive performance of rainfall thresholds for shallow landslides in Switzerland from gridded daily data, doi:10.1002/2017WR021044, 2017.
Peres, D. J. and Cancelliere, A.: Estimating return period of landslide triggering by Monte Carlo simulation, J. Hydrol., doi:10.1016/j.jhydrol.2016.03.036, 2016.
Peruccacci, S., Brunetti, M. T., Gariano, S. L., Melillo, M., Rossi, M. and Guzzetti, F.: Rainfall thresholds for possible landslide occurrence in Italy, Geomorphology, 290, 39–57, doi:10.1016/j.geomorph.2017.03.031, 2017.

AA – Thank you for the suggestion. Following this suggestion and other suggestion coming from Referee1 (reducing Italian references and adding some more suggested references), we performed an update to the references cited in the manuscript.
Please, note that the "rainfall thresholds" topic is very broad, therefore we focused the introduction on literature thresholds that operate into EWS and on thresholds considering antecedent rainfall as a proxy for soil moisture conditions. We were forced to a very strict focus, because the manuscript typology demands a limitation to max. 20 references.
Please, let us know if you believe we left out some relevant references about thresholds using antecedent rainfall or thresholds integrated into EWS.

-

RC – Perhaps the introduction may take into account that the importance of including soil moisture information in landslide triggering thresholds has been stressed by a recent NHESS invited perspective by Bogaard and Greco, 2017.
Bogaard, T. and Greco, R.: Invited perspectives. A hydrological look to precipitation intensity duration thresholds for landslide initiation: proposing hydro-meteorological thresholds, Nat. Hazards Earth Syst. Sci. Discuss., 1–17, doi:10.5194/nhess-2017-241, 2017.

AA – Indeed, the Referee suggests a very interesting article. We made reference to it in the introduction and also in the conclusion, since we believe that our works expands by a small step the classical rainfall threshold approach towards the direction expressed by Bogaard and Greco: instead of using only rainfall, we try to indirectly encompass the hydrology of the territorial units by using soil moisture data. On this regard, we also made reference to a work very recently submitted to the same special issue by Kanli et al. (2017), which shares a similar perspective.
Canli, E., Mergili, M., and Glade, T.: Probabilistic landslide ensemble prediction systems: Lessons to be learned from hydrology, Nat. Hazards Earth Syst. Sci. Discuss., https://doi.org/10.5194/nhess-2017-427, in review, 2017.

We were not aware of these papers when we submitted the first version of our manuscript.

-

RC – Tab. 1: also the number of landslides and true positives and negatives should be shown, and commented in the text

AA – In the revised version of the manuscript, we expanded the table as suggested and provided the necessary comments in the text.
While performing calculations about true positives, we noticed an error: the total number of landslides (hits+missed) was not the same in SIGMA and SIGMA-U. After a thorough check of the used spreadsheet, we identified an error in the formulas: in a few words, 5 TUs were erroneously not included in the calculations for Table 1. We corrected the spreadsheet formulas and links, and we re-calculated the statistics, which now result even better than the mistaken ones reported in the previous version of the manuscript:

"…*false alarms issued at warning level 1, which are negligible, decreased by 8%, while the very important warning level 3 was erroneously issued 11 times instead of 21 (-48%). False alarms at the intermediate warning level 2 were reduced from 287 to 197 (-31%). Missed alarms are reduced as well: while SIGMA missed 88 alarms, SIGMA-U missed 69 alarms (-22%). This corresponds to a total of 134 missed landslides instead of 214 (-37%). Overall, SIGMA-U hits 789 landslides out of 923 (85.5%), outperforming SIGMA that hits 709 landslides (76.8%)."*

We apologize for the error and we express our gratitude to the Referee that made it possible to notice it and to correct it.

-

RC – Technical corrections
P1 L12 maybe replace "were" with "are"
P1 L22 "thresholds" instead of "threshold"
P1 L23 "landslide occurrence" instead of "the landslide occurrence"
P1 L25 "as a proxy" instead of "a proxy"
P2 L2 "landslide" instead of "the landslide"
P2 L29 here introduce the acronym MSM
P2 L27 "rainfall-runoff" instead of "inflow-outflow"
P4 L14 "importantly" instead of "important"
P4 L29 "is by large" maybe can be improved

AA – All suggested corrections have been included in the revised text.

-
RC – Fig. 1: Where soil punctual measurements were taken?

AA – As explained before, we do not use measurements. In the manuscript, we use the term "punctual estimates". Values are not actually measured, they are estimated by TOPKAPI model. We use "punctual" to stress that we are not performing a distributed assessment [e.g. on a pixel-by-pixel basis]: since the original EWS uses only a rainfall measuring station for each territorial unit, we need only a soil moisture value for each territorial unit.

In the conclusions section, we only hypothesize the possibility of using actual measures in the future developments of the research (of course, provided the funds are renewed and the research plan is approved).

Maybe in the introduction a sentence could be misleading. To avoid misunderstandings, it was changed.
PREVIOUS VERSION OF THE TEXT
*"This work explores the possibility to exploit punctual soil moisture values estimated at few discrete points and to correlate them with the landslide triggering over wide areas (thousands of squared kilometers)".*

REVISED TEXT
*"This work explores the possibility to exploit the estimated mean soil moisture value averaged over large (thousands of squared kilometers) territorial units to find an empirical correlation with landslides triggering"*

A similar modification was performed in the conclusion

PREVIOUS TEXT
*We improved a state of the art RSLEWS based on rainfall thresholds (SIGMA, Martelloni et al., 2012; Lagomarsino et al., 2013) by integrating punctual soil moisture estimates.*

REVISED TEXT
*We improved a state of the art RSLEWS based on rainfall thresholds (SIGMA, Martelloni et al., 2012; Lagomarsino et al., 2013) by integrating mean soil moisture values averaged over the territorial units of the system.*

-

RC – Fig. 2: on the upper-left: there must be a mistake in the orientation of the arrows

AA – Thank you for identifying this error. The figure was adjusted.

[revised manuscript text omitted]

---

## Author Response (AR2)

Dear Editor, thank you for accepting our work for publication.

All suggestions from reviewer have been addressed in the new version of the manuscript.

We would like to draw your attention on an additional change we performed: WE SLIGHTLY CHANGED THE TITLE OF THE MANUSCRIPT.

FORMER TITLE:

Brief communication: Using **PUNCTUAL** soil moisture estimates to improve the performances of a regional scale landslide early warning system

CURRENT TITLE:

Brief communication: Using **AVERAGED** soil moisture estimates to improve the performances of a regional scale landslide early warning system

REASON:

During the revision process, referees and authors concurred that the term "punctual" was misleading. It was substituted with "averaged" throughout the text. The only occurrence left for the word "punctual" was in the title; therefore, we decided to modify it.